# Spontaneous Polarization Calculations in Wurtzite II-Oxides, III-Nitrides, and SiC Polytypes through Net Dipole Moments and the Effects of Nanoscale Layering

**DOI:** 10.3390/nano11081956

**Published:** 2021-07-29

**Authors:** William Troy, Mitra Dutta, Michael Stroscio

**Affiliations:** 1Department of Electrical and Computer Engineering, University of IL at Chicago, Chicago, IL 60517, USA; dutta@uic.edu (M.D.); stroscio@uic.edu (M.S.); 2Physics Department, University of IL at Chicago, Chicago, IL 60517, USA; 3Richard and Loan Hill Department of Biomedical Engineering, University of IL at Chicago, Chicago, IL 60517, USA

**Keywords:** spontaneous polarization, wurtzite, nanolayer, dipole, SiC, GaN, semiconductor, built-in electric field, hexagonal lattice

## Abstract

Herein, the spontaneous polarization in crystals with hexagonal symmetry are calculated as a function of the number of monolayers composing a nanostructure by adding the dipole moments for consecutive units of the nanostructure. It is shown that in the limit of a large numbers of monolayers that the spontaneous polarization saturates to the expected bulk value of the spontaneous polarization. These results are relevant to understanding the role of the built-in spontaneous polarizations in a variety of nanostructures since these built-in polarizations are generally quite large, on the order of 1 × 10^8^ to 1 × 10^10^ V/m. Using these formulations, we come to the prediction that small nanolayered structures are theoretically capable of having larger spontaneous polarizations than their bulk counterparts due to how the dipole moments of the anions and cations within a wurtzite lattice cancel out with one another more in larger structures.

## 1. Introduction

Spontaneous polarization, P_sp_, is a polarization in materials that have asymmetric charge distributions, notably in wurtzite structures, and has been verified by both experiments and in highly complex modeling techniques such as Bloch’s Theorem in Density Functional Theory and ab initio Berry-phase calculations for periodic systems [1,2]. These studies of spontaneous polarization are promising for development of multifunctional surfaces, structures, and thin films with superior properties [3]. In this work we seek to treat finite systems—that are not assumed to be periodic—by adding the dipole moments of successive layers to determine how the spontaneous changes with the number of monolayers until the spontaneous polarization saturates to the bulk value. That is, here we show that the bulk P_sp_ can be calculated as a net dipole moment over volume based on Equation (1); the primitive unit cell is shown in Figure 1. With the bulk P_sp_ values of wurtzite II_B_-VI, II_B_ referring to elements in the zinc family while II_A_ are alkaline earth metals, semiconductors such as CdSe, CdS, and ZnO being accurately calculated as primitive lattice net dipole moments [4].
(1)$$P_{\mathrm{sp}}=\frac{1}{V}\sum e_{i}^{*}d_{i}$$

Here, we simplify the equations of [5] to a singular equation that can be used to calculate the P_sp_ of wurtzite materials before extending these calculations to III-Nitrides as well as H-SiC polytypes to confirm the validity of this approach to new materials. Then, we extend this approach to calculate the P_sp_ values of nanoscale quantum well or thin film structures consisting of differing numbers of monolayers which differ from that of bulk. The underlying causes for P_sp_ to differ for such nanoscale structures and for the bulk is non-integer number of elementary lattices within the overall structure and surface/interface charges, an effect which is negligible in bulk values but can have a large effect on the P_sp_ values in small nanoscale lattices [6,7,8]. This effect of the P_sp_ deviation of nanostructures from that of bulk and the multiplication of this effect due to how the bonds of the surface atom terminate will become clear both in theory and mathematically as we continue. 

## 2. Materials and Methods

As described in [4] in the Modern Theory of Polarization the overall polarization of a structure can be modeled as the net sum of dipole moments within the structure. This theory was built upon by [5] using bond vector addition to get the permanent electric dipole moment, $\mu_{0}$, of the elementary lattice cell, Equation (2), and taking the volume of the elementary unit cell, v, Equation (3), to get the overall polarization of the structure as $P_{\mathrm{sp}}=\mu_{0}/v_{0}$.
(2)μ0=e*(u−38)c=e*4(c1+3c2cos(ϑ)) 
(3)$$v_{0}=\frac{\sqrt{3}a^{2}c}{4}\;$$

Using Equations (2) and (3) we combine them to get Equation (4), which calculates the P_sp_ values of a wurtzite elementary unit cell from its elementary lattice parameters.
(4)$$P_{\mathrm{sp}}=\frac{e^{*}\left(4u-1.5\right)}{\sqrt{3}a^{2}}=\frac{e^{*}\left(3c_{2}-c_{1}\right)}{2\sqrt{3}a^{2}\left(c_{1}+c_{2}\right)}\;$$
where $e^{*}$ is the effective charge of an atom in a binary lattice, taken to be the Born effective charge, a is a lattice constant, $c_{1}$ is the anion bond length with the northern cation given by $c_{1}=uc$, and $c_{2}$ is the anion bond length with the southern cations (C_2_-C_4_) $c_{2}=-cos\left(\vartheta \right)b=\frac{c}{2}-c_{1}$ in the $\hat{c}$ direction; see Figure 2.

It should be noted that the Born effective charge differs from that of the formal ionic charges of the atoms and from the Modern Theory of Polarization its general form is given by Equation (5) [4], which represents the change in polarization as a result of the displacement of an atom.
(5)$$e_{ij}^{*}=\frac{\Omega }{e}\frac{\delta P_{i}}{\delta d_{j}}\;$$

From Equation (4) it can be seen that in the ideal wurtzite structure where *u* = 0.375, ϑ = 109.47°, and *c*/*a* = $\frac{2\sqrt{6}}{3}$, no matter the effective charge, there is no P_sp_. This is caused by a cancellation of net dipole moments from the negative dipole moment of the C_1_-A_1_ cation-anion bond with the 3 positive southern cation-anion dipole moments, in all wurtzite structures *C*_1_ must be greater than 3 × *C*_2_, as seen in the numerator of Equation (2). This is because all known wurzites have negative P_sp_ values and results in P_sp_ being highly dependent on the cell-internal structural parameter, *u*, specifically.

Using Equation (4) and known lattice parameters of various binary pure wurtzite structures we can calculate their bulk P_sp_ values and compare against that of both experimental and theoretical results, Table 1.

From here we can separate out Equation (4) into separate dipole moments and lattice volumes of the top C_1_-A_1_ bond and the bottom 3 cation-anion bonds as Equations (6)–(9).
(6)$$u_{1}=-\frac{e^{*}c_{1}}{4}\;$$
(7)$$u_{2}=\frac{3e^{*}c_{2}}{4}\;$$
(8)$$v_{1}=\frac{\sqrt{3}a^{2}c_{1}}{2}\;$$
(9)$$v_{2}=\frac{\sqrt{3}a^{2}c_{2}}{2}\;$$
where *u* is the dipole moment in C-m and *v* is the volume in m^3^. At this point, one can use *u_i_* and *v_i_* to calculate the sheet charge density as $\sigma_{i}=u_{i}/v_{i}$, in C/m^2^, and the charge per unit length in the $\hat{c}$ direction as $P_{i}=\sigma_{i}c_{i}$, in C/m. Treating each bond layer in a wurtzite lattice as charges per unit length which are functions of the atomic dipole moments we can now express P_sp_ as the sum of *Pi* values over a given distance divided by the total distance, in the $\hat{c}$ direction, Equation (10).
(10)$$P_{\mathrm{sp}}=\frac{\sum P_{i}}{\sum c_{i}}\;$$

If we then think of a layer of a lattice as an elementary unit cell, given by Figure 2, we can see that even as we separate out the dipole moments within the structure the net dipole moment of the elementary unit cell is equal to that of bulk. This is the case as if we use Figure 2 as an example the 3 bottom cations individually have 3 positive, equal, dipole moments in the c-direction with the center anion. Just as the bottom 3 cations have individual dipole moments with the center anion the top cation has its own dipole moment with the center anion, except in the negative direction, and due to the further distance with respect to the c-axis between the top cation and center anion, when compared the $\hat{c}$ distance between any bottom cation with the center anion, this dipole moment is much stronger. This stronger top dipole moment over the sum of the bottom 3 dipole moments leads to a net negative dipole moment of a wurtzite which is equal to that of bulk.

## 3. Results

Using this methodology, we show that the elementary unit cell has the P_sp_ value as that of bulk, Table 2. Using the equations above we can then convert this to sheet charges. These sheet charges can be defined as C and A for anion and cation and as long as they are periodic about their dipole moments and distance, they will retain the P_sp_ value of that of bulk as one can think of P_sp_ as similar to density in that it is the net dipole moment per unit length vector. In Figure 3 a single period, or layer, of the structure can be defined as either A-C-A, Figure 3a, or C-A-C, Figure 3b, these two periods are equal in their P_sp_ values as long as we are working with a wurtzite lattice or any other periodic structure. As we add more periodic layers to the structure such as A-C-A-C-A or C-A-C-A-C as stated the structure still retains the P_sp_ of that of its bulk. However, if we break this periodicity and introduce partial period layers such as C-A-C-A-C-A or A-C-A-C-A-C we now have a structure that has a P_sp_ that differs from that of its bulk value.

### 3.1. Purely Hexagonal Lattice Structures

This change in the P_sp_ from bulk caused by having a non-periodic layer while the rest of the structure is periodic is illustrated in Figure 4, where the overall P_sp_ is given for various wurzites as a function of the number of periodic layers within it but starting with only a half layer, this layering structure being shown in Figure 3.

As we can see from Figure 4 even though the P_sp_ of the layered structure is initially dominated by the non-periodic layer the P_sp_ saturates out as we add more periodic layers to that of bulk. This is because P_sp_ as given by Equation (10) is the average dipole moment per unit length vector.

In keeping conservation of charge, we treat the atoms of surface layers as having a charge of $e^{*}$/4. This is the case as if we think of in Figure 1 the C_1_ atom being the surface atom there is only 1 bond connecting it to the rest of the structure, the C_1_-A_1_ bond. Meaning that the charge donation to this atom is only ¼ what it would be compared to if we took a non-surface atom from within the structure that has 4 bonds connected to it, each of these 4 bonds having a charge of $e^{*}$/4, leading to a total charge on a non-surface atom of $e^{*}$. Even though the surface atom’s charge is only ¼ that of its inner layer counterparts we can assume its bond retains the charge of $e^{*}$/4 as it does not have to split it’s charge with other bonds and the charge on the atom is donated through this bond as well, leading to a majority of the valence electron density of this atom being in the area occupied by its bond as the charge on the atom comes from this bond as well.

It should also be considered that the number of layers directly corresponds to the thickness of the semiconductor as each periodic layer is equal to c/2. Meaning in the case of AlN even if we take the more conservative estimate of Figure 5 and say that it takes approximately 50 layers for the P_sp_ of the structure to be considered that of bulk that is 15.6 nm thick layer and in the case of quantum dots, quantum wells, or other nanoscale structures such as layers in a MOSFET this can be non-negligible. This result is also very interesting as it leads to a non-intuitive effect in which smaller structures are able to create large spontaneous polarization induced electric fields surrounding them, up to and greater than 10^10^ V/m in these cases, which for many applications is quite massive.

It should also be noted with these equations that the atom in this surface layer which we treat as non-periodic can be easily replaced with an atom not native to the overall periodic wurtzite structure such as if one deposited an atomically thin layer of an element on top of the wurtzite structure through chemical vapor deposition or some other means as long as the bond length and effective charge transfer of this layer with respect to the top of the overall periodic structure is known.

### 3.2. Mixed Cubic/Hexagonal Lattice Structured SiC 

In proving the validity of our model in simplistic purely hexagonal wurtzite structures can now further extend out model to the more complex partially hexagonal structures of 4H- and 6H-SiC. These polytypes are special in that their lattices are partially cubic and partially hexagonal, 50% and 33%, respectively. As discussed, while bulk cubic lattices retain dipole moments between atomic bonds within the structure, they are nonpolar structures overall, this is caused by the cancellation of their atomic dipole moments. Since 4H- and 6H-SiC are only partially cubic they still have a bulk P_sp_. Furthermore, we observe than although in normal cubic structures there is no polarization, there is a slight net polarization in the cubic regions of the 4H- and 6H-SiC structures due to their cubic bonds being shifted from that of bulk, resulting in dipole moments. This can be thought of as similar to how BaTiO_3_ in its normal cubic structure exhibits no P_sp_ but when below its Curie temperature its Ti and O atoms shift, causing it to become a polar structure. Except in the case of SiC this atom shift is not caused by temperature but instead it is caused by the stress on the cubic Si and C atoms from the hexagonal Si and C atoms. 

Exact atom positions can be calculated by methods given in [17] with calculated 4H- and 6H-SiC Si and C atom bond lengths relative to the c-axis in Figure 5 shown in Table 2.

From the atom bond lengths in Table 2, Born effective charge of SiC in Table 1, and a values of 3.08051 and 3.08129 Å we calculate the bulk P_sp_ of 4H- and 6H-SiC of −0.00568 C/m^2^ and −0.00452 C/m^2^, respectively. These calculated values are well within the range of other’s theoretical values of −0.0055 to −0.0168 C/m^2^ for 4H-SiC and −0.0036 to −0.0111 C/m^2^ for 6H-SiC [16]. It should be said that due to the shift in the cubic atom positions, due to the strain on them from the hexagonal atoms, their dipole moments do not cancel out as nicely as they do in a traditional fully cubic lattice. This makes it so calculation of the P_sp_ for partially hexagonal SiC lattices, at least in cases considering net dipole moments, is not as simple as just multiplying the P_sp_ of a pure hexagonal lattice by the percentage of hexagonality as other techniques for calculating Psp of mixed lattice structures have tried to do [16] If this was the case, we would have gotten P_sp_ values of −0.014305 C/m^2^ and −0.0095367 C/m^2^.

Using the same methodology for purely hexagonal wurtzite structures we calculate the Psp of the partially hexagonal 4H- and 6H-SiC lattices as a function of the number of layers, Figure 6.

In the cases of 4H- and 6H-SiC there are 4 and 6 possible different surface terminations. For 4H-SiC these possible surface terminations are the Si(1), C(1), Si(2), and C(2) atoms, with each periodic layer being made up of Si(1)-C(1)-Si(2)-C(2), repeating for n number of layers. However, what we define as a “periodic layer” depends upon the surface termination, the previous example given was for a C(2) surface termination. If we instead say that the surface terminates at a Si(2) atom then each periodic layer is made up of C(2)-Si(1)-C(1)-Si(2) bonds in that order. No matter how we define the “layer” for 4H-SiC as long as all 4 atoms are in it there is the same polarization of the layer, equal to that of bulk. One can easily see how this layering structure extends to 6H-SiC with its Si(1), C(1), Si(2), C(2), Si(3), and C(3) atom structures.

## 4. Conclusions

In this work we demonstrate that using a simple calculation of net dipole moments we are able to accurately calculate overall sponantanous polarization of various purely hexagonal and partially hexagonal wurtzite structures. We show that this methodology while very simplistic when compared to its highly complex Berry Phase and Bloch’s Theorem counterparts, or experimental techniques for measuring spontaneous polarization of structures, is just as accurate, if not more in certain cases, while being much more user friendly and not hiding behind what is the “black box” of density functional theory to many. From this initial methodology we then extend it to investigate the effect of the surface termination in nanoscale systems. As This investigation leads to the realization that the polarization of nanostructures deviates quite profoundly from that of bulk is highly dependant on the number of layers of the structure and its surface terminating atoms, something that is rarely investigated or taken into account. This effect can have large impacts on a wide range of nanostructured systems from nanoparticles using their natural spontaneous polarization to open ion channels [18], layering in a MOSFET and creation of spontaneous polarization induced 2D electron-gasses [19], quantum well based structures such as quantum cascade lasers [20], or quantum dot solar cells [21]. It would be interesting to see how this nanolayering effect can be expanded to higher order structures [22] and how these equations can be expanded upon to more accurately model nonuniform objects such as nanospheres, nanostars, and nanorods. Not only this but from these equations one may be able to introduce a polarization in the direction perpendicular to $\hat{c}$, which normally does not exist in wurtzite structures due to lattice symmetry but is theoretically possible to be non-zero in nanostructured systems based on our presented equations.

It should also be noted that with these equations the structures do not necessarily need to have periodic dipole moments to calculate the Psp of the structure. All that is needed are the atom positions, or bond lengths with respect to a predefined axis, and the Born effective charge of the atoms in each particular lattice. From these things one can calculate the Psp of any atomic structure with regard to any defined axis, allowing for a surprisingly intuitive and simplistic calculation for what has been considered to be very complex in nature.

## Figures and Tables

**Figure 1 nanomaterials-11-01956-f001:**
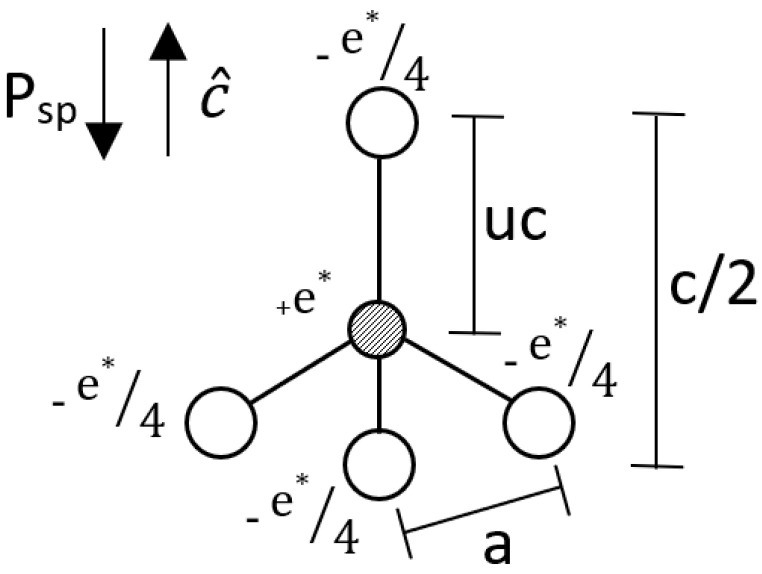
Point charge model of a elementary wurtzite unit cell along with lattice parameters.

**Figure 2 nanomaterials-11-01956-f002:**
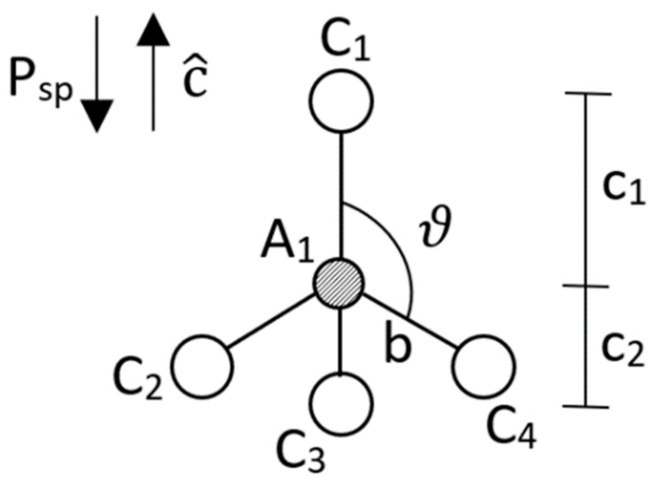
Wurtzite elementary unit cell where atoms are labeled as anions, A, and cations, C, and the parameters b and ϑ being the bond length between the A_1_ atom and any of the southern cations (C_2_–C_4_) and the bond angle between the C_1_ atom and any of the southern cations (C_2_–C_4_), respectively.

**Figure 3 nanomaterials-11-01956-f003:**
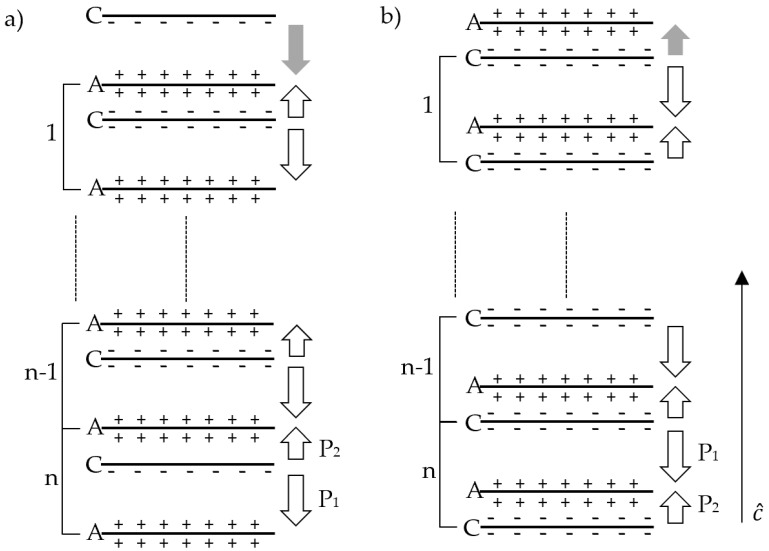
Depiction of layering structure where A is the anion and C is the cation of a binary wurtzite structure in the c-direction. (**a**) is case 1 where there is an extra cation at the surface that is not cancelled out with each “layer” consisting of an A-C-A bond. (**b**) represents the other possible case, case 2, where there is an extra anion at the surface that is not cancelled out. Each layer in case b consists of a C-A-C bond. In the case of SiC where it is not easy intuitive as to which is the anion and which is the cation, Si is the cation and C is the anion.

**Figure 4 nanomaterials-11-01956-f004:**
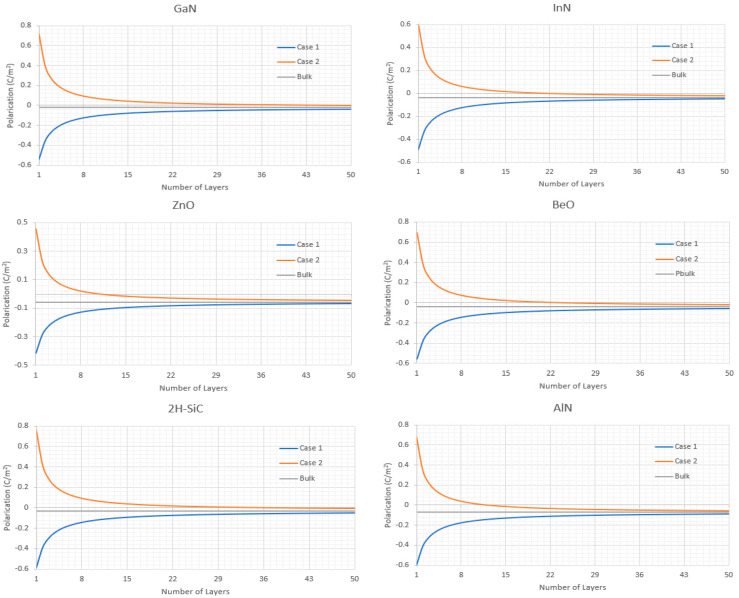
P_sp_ as a function of the number of atomic layers for various pure hexagonal wurtzite structures.

**Figure 5 nanomaterials-11-01956-f005:**
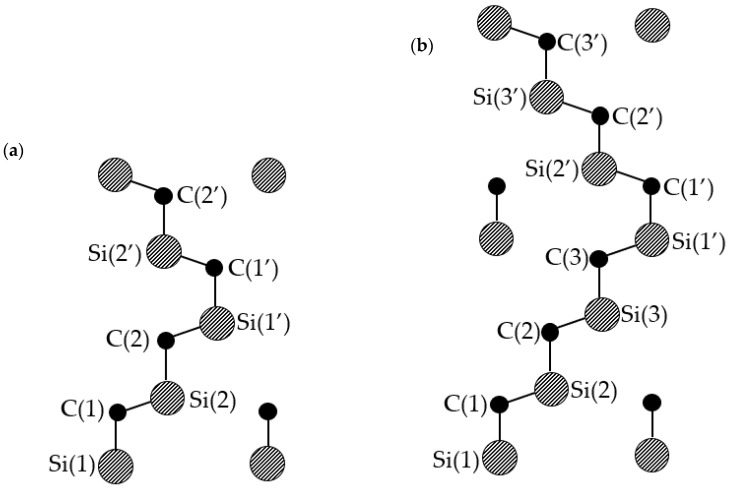
Lattice structures of (**a**) 4H-SiC and (**b**) 6H-SiC. Where in both cases the Si(1)-C(1) and Si(1′)-C(1′) bonds are hexagonal and the rest are cubic in nature. The C atoms are solid black and the Si atoms are hashed.

**Figure 6 nanomaterials-11-01956-f006:**
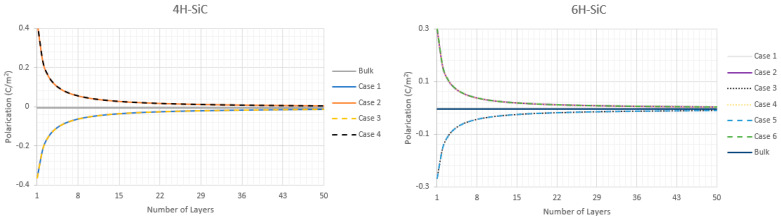
P_sp_ as a function of the number of periodic layers for 4H-SiC and 6H-SiC.

**Table 1 nanomaterials-11-01956-t001:** Experimentally obtained lattice parameters for various wurtzites along with Born effective charges and P_sp_ values which were obtained both experimentally and theoretically.

Wurtzite	a (Å)	c (Å)	u	$e^{*}\;(e)$	P_sp_ Calculated (C/m^2^)	P_sp_ Others (C/m^2^)
ZnO	3.25 [9]	5.206 [9]	0.3819 [9]	2.05 [10]	−0.04954	−0.057 [10]
BeO	2.688 [9]	4.351 [9]	0.379 [9]	1.943 [9]	−0.0398	−0.0363 [9]
InN	3.585 [11]	5.80053 [11]	0.379 [11]	3.02 [10]	−0.03477	−0.032 [10]
AlN	3.11 [11]	4.9947 [11]	0.382 [11]	2.7 [10]	−0.07229	−0.040 to −0.081 [10]
GaN	3.192 [12]	5.185 [12]	0.377 [12]	2.72 [10]	−0.01975	−0.018 to −0.23 [13]
2H-SiC	3.079 [14]	5.11046 [14]	0.3777 [14]	2.7 [15]	−0.02861	−0.0111 to −0.0432 [16]

**Table 2 nanomaterials-11-01956-t002:** Distances between Si and C layers in wurtzite 4H-SiC and 6H-SiC lattices in angstroms relative to the c-axis in Figure 6.

	Si(1)-C(1)	C(1)-Si(2)	Si(2)-C(2)	C(2)-Si(1′)	C(2)-Si(3)	Si(3)-C(3)	C(3)-Si(1′)
4H-SiC	1.89695088	0.624047424	1.89029492	0.631107			
6H-SiC	1.89556	0.624849	1.89042		0.627268	1.89042	0.631351

## Data Availability

All data used and/or analyzed during the current study are shown within the study. If further data is required, it can be made available to the corresponding author upon reasonable request.

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
