# Peer review of "Spontaneous Polarization Calculations in Wurtzite II-Oxides, III-Nitrides, and SiC Polytypes through Net Dipole Moments and the Effects of Nanoscale Layering"

_nanomaterials, 2021, doi:10.3390/nano11081956_

Round 1

Reviewer 1 Report

This manuscript contains interesting approach to demonstrate polarization. The text is comprehensive, but I recommend to follow Authors rules of the Journal for presentation of the material (it means to divide the manuscript to the recommended chapters). Some information about importance of the study belongs rather to Introduction part.  Please, support this sentence by additional literature: “It would be interesting to see how this nanolayering effect can be expanded to higher order structures [DOI: 10.1016/J.TSF.2012.11.023] and how these equations can be expanded upon to more accurately model nonuniform objects such as nanospheres, nanostars, and nanorods [.... please look for corresponding ref.].” I suggest to add the following statement “Studies of spontaneous polarization are promising for development of multifunctional surfaces, structures and thin films with superior properties [DOI: 10.3390/NANO10101990]”

Reviewer 2 Report

This manuscript could be accepted for publication in Nanomaterials. The novelty of presented research and it's importance for the scientific community are high. The introduction provide sufficient background and include all relevant references. The research methodology is adequate. The results are clearly presented. The conclusions supported by the data. The manuscript good illustrated and interesting to read. English language and style are fine. Probably, the only point which should be fixed: the manuscript should be prepared in the journal's template.

Reviewer 3 Report

In this manuscript, the authors considered finite systems that are not assumed to be periodic, by adding the dipole moments of successive layers to determine how spontaneous polarization changes with the number of monolayers until it reaches the bulk value. This study leads to the realization that the polarization of nanostructures is quite different from that of bulk and strongly depends on the number of layers of the structure. This phenomenon is rarely investigated or taken into account. However, the manuscript needs a thorough editing; taking into account the comments below, before I could recommend its publication.
1. The physical meaning of the last sentence in the abstract is not clear. Why can spontaneous polarizations of small nanolayered structures be greater than their bulk analogs?
2. Section 2 should be carefully edited.
(i) The designations of physical quantities should be written in a uniform manner, for example, in italic type.
(ii) All the variables in Fig. 2 should be defined either in the text or in the figure caption.
(iii) First paragraph on page 3.
“… in all wurtzite structures b1 must be greater than 3x b2, as seen in the numerator of eq. 2.” In equation (2) there are no variables b1 and b2. These variables are not defined in the text.
(iv) According to eq. (4), Psp is expressed in terms of c1, c2 and u. These quantities, in turn, depend on b1 and b2.
3. The last paragraph is in the Conclusion.
The physical meaning of the term “the effective charges of bonds” is not clear.
